# Evaluation of Bilateral Maxillary Sinus Ectopic Teeth Using CT and Cinematic Rendering—A Case Report

**DOI:** 10.3390/diagnostics13193084

**Published:** 2023-09-28

**Authors:** Dario Baldi, Liberatore Tramontano, Bruna Punzo, Carlo Cavaliere

**Affiliations:** IRCCS SYNLAB SDN, Via Emanuele Gianturco 113, 80143 Naples, Italy; dario.baldi@synlab.it (D.B.); liberatore.tramontano@synlab.it (L.T.); carlo.cavaliere@synlab.it (C.C.)

**Keywords:** ectopic teeth, CT, cinematic rendering, maxillary sinus, case report

## Abstract

Ectopic teeth in the maxillary sinus are a rare finding and pose a diagnostic challenge due to their unusual location and clinical management. A 28-year-old man presented with complaints of discomfort and pressure in the maxillary sinus region. A CT scan and cinematic rendering revealed the presence of ectopic teeth in the maxillary sinus bilaterally. The use of cinematic rendering provided a more detailed and accurate visualization of the ectopic teeth and surrounding anatomical structures. A CT scan is the primary imaging modality used for the diagnosis and visualization of ectopic teeth in the maxillary sinus. In addition, the use of cinematic rendering can improve diagnostic accuracy and reduce the need for further imaging studies. The use of CT and cinematic rendering can help in the diagnosis and visualization of ectopic teeth in the maxillary sinus, aiding in the planning of surgical interventions.

## 1. Introduction

Ectopic eruption of teeth is a rare but important condition in dentistry. It is a developmental disorder that can cause teeth to emerge from locations outside of their normal arch or bone boundaries. Indeed, the development of teeth begins early in life with the formation of the dental lamina. Abnormal interactions between the oral epithelium and the underlying mesenchymal tissue can lead to ectopic tooth development and eruption [1].

Although ectopic teeth can emerge from various parts of the body, including the palate, mandibular condyle, coronoid process, orbits, nasal cavity, or even through the skin, the maxillary sinus is one of the rarest and most clinically significant sites for this finding. Due to the rarity of this condition, its incidence deserves to be added to the literature and discussed. Teeth ectopically erupted in the maxillary sinus can cause sinusitis, ocular symptoms, and other complications, and the condition may go undiagnosed for years until the patient undergoes radiography for another reason [2].

In some cases, ectopic teeth in the maxillary sinus may be associated with dentigerous cysts, which are benign odontogenic cysts that develop from the crown of impacted or unerupted teeth. Dentigerous cysts are thought to originate from the follicle of the unerupted tooth and are caused by the expansion of the dental follicle due to the accumulation of fluid between the tooth crown and epithelial components. The presence of cysts and ectopic teeth can cause a range of complications, from sinus obstruction to blindness.

The identification of ectopic teeth in the maxillary sinus may have implications for preoperative planning in cases of maxillary sinus pathology, particularly if surgery is required. Advanced imaging techniques such as computed tomography (CT) and magnetic resonance imaging (MRI) can provide improved characterization of different bone/soft tissues and their relationships. In addition, post-processing techniques, such as cinematic rendering (CR), represent an advanced 3D visualization technique, particularly useful to detail complex anatomical regions and also for presurgical planning [3].

## 2. Case Report

### 2.1. Case Presentation

We present a case of a 28-year-old man who was referred to our institution with a complaint of hypertrophy of the lymphoid tissue in the nasopharynx and turbinates. The patient reported symptoms of nasal obstruction, rhinorrhea, and occasional epistaxis. There was no significant medical history or previous surgical interventions. The patient gave written informed consent for this study.

### 2.2. Materials and Methods

The patient underwent a CT scan using a Siemens Somatom Force DSCT scanner (Siemens Healthineers, Erlangen, Germany). The maxillo-facial CT scan was performed without contrast agent injection using the following parameters: 120 kVp, 100 mAs, a slice thickness of 0.625 mm, and a 0.3 mm gap. The images were reconstructed using a soft tissue algorithm (head regular (HR) 64; Advanced Modeled Iterative Reconstruction (ADMIRE) strength level: 3).

The CT images were reviewed and reported by a radiologist with expertise in interpreting head and neck CT scans. In addition, a specialized radiologic technologist in 3D imaging performed CR of the entire data set using Cinematic Playground (Siemens Syngo.Via; Siemens Healthineers, Erlangen, Germany), a specialized software (software version vb60a hf04) that generates CR from CT and MR acquisitions. The radiologist evaluated the presence of any anatomical variations or abnormalities in the paranasal sinuses and nasal cavity (Figure 1, Figure 2 and Figure 3).

## 3. Discussion

An impacted tooth can get stuck in gum tissue or bone for a number of reasons. The area may be overcrowded, leaving no room for the tooth to emerge. For example, the jaw may be too small to hold the wisdom teeth. Teeth can also become twisted, tilted, or displaced as they try to come out.

The tooth, which fails to erupt in the oral cavity in its functional position, has lost its further potential for eruption according to Archer’s classification (1975) for the imposition of maxillary third molars [4].

The maxillary third molar is not impacted as frequently as the mandibular third molar, but the frequency of impaction is high. Impaction of the maxillary third molar is considerably more difficult due to many factors that are favorable compared to the mandibular third molar, such as the visibility factor, reach, tuberosity, and its proximity to the maxillary sinus. These are the many factors that make maxillary impaction a challenge for dentists.

The different types of maxillary third molar impingement are similar to mandibular third molar impingement based on the angulation, prevalence, and prediction of extraction difficulty of the tooth in relation to the second molar. Archer qualitatively classified them into seven types: mesioangular (43%, most common), distoangular, vertical, horizontal (about 3%, less common), buccoangular, linguoangular, and inverted [4].

Several theories have been proposed to explain the occurrence of ectopic teeth in the maxillary sinus, including migration of tooth buds, developmental abnormalities, trauma, and infection. Diagnosis can be challenging, considering that most ectopic teeth have no symptoms. In spite of this, impacted teeth can cause orofacial pain due to their location, mimic temporomandibular joint pain, be a source of focal infection, or develop an odontogenic cyst or odontogenic tumor in conjunction with follicle dysplasia due to the pathology. Imaging techniques such as maxillo-facial CT and panoramic radiographs are often used to aid diagnosis.

Maxillary third molars rank second in terms of teeth most commonly impacted, while mandibular molars top the list. Since the third molar is the final tooth to emerge in the maxilla, it is more prone to displacement due to limited space, potentially explaining the increased occurrence of ectopic third molars in the maxilla. The genesis of teeth, or odontogenesis, starts during the sixth week of fetal development, coinciding with the formation of the dental lamina in both the maxilla and mandible [5]. The formation of a mature tooth, including its crown and root, arises from intricate tissue interactions between the oral epithelium and the mesenchyme beneath it. Disruptions or abnormalities in these interactions can result in the development and emergence of ectopic teeth. Typically, these ectopic teeth are identified in individuals in their twenties or thirties. The observed age range spans from 4 to 57 years, averaging around 28.06 years. Notably, males have a two-fold higher likelihood of experiencing this compared to females [6].

The first report of an ectopic maxillary third molar was published in 1952 by Isasi Garcia et al. [7]. Ectopic third molars in the maxilla represent 30% of all such occurrences, while the mandible accounts for the remaining 70%. In adults, these ectopic molars often manifest clinical symptoms like pain, swelling, or abscesses. Occasionally, they might be detected unexpectedly during standard radiological scans. An unrestricted literature search was performed, and only five cases were reported as bilateral ectopia [8,9,10,11,12].

Daimi et al. (2020) presented a case study involving a dry human skull with a bilateral ectopic third maxillary tooth. The estimated age range of the skull was between 30 and 35 years. The tooth was found to be partially erupted on both sides from the infratemporal surface of the maxilla, displaying a downward and lateral direction of eruption. Morphometric measurements were conducted to assess the relationship between the tooth and surrounding anatomical landmarks, including the midsagittal plane, alveolar canal, pterygomaxillary fissure, and superior alveolar margin. These measurements provided valuable insights into the position and orientation of the ectopic tooth within the skull [8].

In a separate case reported by Arici et al. (2022), a 32-year-old female patient presented with bilateral ectopic maxillary third molars situated within the maxillary sinuses. The patient experienced symptoms such as facial swelling, pus discharge from the nostril, and nasal airway obstruction. A surgical intervention was performed, involving the removal of the impacted third molars and the enucleation of the associated dentigerous cyst. The procedure successfully alleviated the patient’s symptoms, and a one-year follow-up confirmed the positive outcome [9].

Similarly, Sharma et al. (2019) presented a case of a 27-year-old female patient with recurrent facial swelling and purulent nasal discharge. Clinical examination revealed mild facial swelling and tenderness, along with missing maxillary third molars. Radiographic evaluations, including a panoramic radiograph and cone-beam computed tomography (CBCT), confirmed the presence of ectopic maxillary third molars within the maxillary sinuses. To address the condition, surgical removal of all impacted third molars and enucleation of the associated dentigerous cyst were performed under general anesthesia. The patient experienced complete resolution of symptoms, and histopathological examination confirmed the absence of malignancy [10].

In a different context, Berberi et al. (2023) presented a case involving a 24-year-old female patient referred for the assessment of her third molars prior to orthodontic treatment. A clinical examination revealed minor pain in the anterior and posterior maxillas, prompting a radiographic evaluation. Both 2D panoramic radiography and CBCT confirmed the presence of bilateral ectopic maxillary third molars located within the maxillary sinus cavities. The findings were crucial in determining the treatment approach before proceeding with orthodontic intervention [11].

Lastly, Al Khudair et al. (2019) documented the case of a 19-year-old male patient with complaints of facial pain over the upper jaw and post-nasal discharge. The patient reported recurring sinusitis episodes but lacked any other noteworthy medical history. A CT scan of the paranasal sinuses uncovered bilateral cystic formations and ectopic teeth in both maxillary sinuses. The patient then received an endonasal endoscopic procedure to remove the cysts and extract the impacted teeth. Subsequent post-operative assessments verified the alleviation of symptoms [12].

These cases highlight various presentations of ectopic maxillary teeth within the maxillary sinuses and emphasize the importance of appropriate diagnostic imaging and surgical intervention to address associated symptoms and prevent complications.

None of the reported works and cases have ever used volume rendering (VR) or CR for three-dimensional visualization of the molar in the maxillary cavity.

Ectopic teeth are relatively common, although they are usually asymptomatic. However, when an ectopic tooth is associated with a dentigerous cyst and invades the maxillary sinus, symptoms may present late, including facial pain, swelling, headaches, purulent discharge, and nasolacrimal obstruction. In our patient’s case, his main complaint was pain when blinking his right eye, as well as headache and maxillary pain, often accompanied by postnasal discharge and recurrent sinusitis.

The effects of an ectopic tooth in the maxillary sinus can include facial fullness, headaches, recurrent chronic sinusitis, local sinonasal symptoms, and elevation of the orbital floor. If the lesion extends along the floor of the orbit, it can cause diplopia and even blindness.

In such cases, a thorough clinical examination and diagnostic imaging, such as maxillo-facial CT, are essential to accurately diagnose and monitor the condition.

CT provides detailed information about the location and morphology of the ectopic tooth, as well as the evaluation of surrounding anatomical structures, facilitating treatment planning.

In the present case, CT was used to visualize the ectopic teeth in the maxillary sinus. CT has been widely used to diagnose ectopic teeth in the maxillary sinus because it can provide detailed information about the location and morphology of the ectopic tooth. In addition, CT allows evaluation of the surrounding anatomical structures, such as the sinus walls and adjacent teeth, in light of a possible surgical treatment, facilitating its planning [13].

We also used cinematic rendering to visualize the ectopic teeth in the maxillary sinus. By simulating the propagation and interaction of light beams as they move through the volumetric data, cinematic rendering creates a representation of 3D pictures that is more lifelike than that produced by ordinary volume rendering [11]. In general, cinematic rendering uses the same procedures as volume rendering to determine color and opacity: transfer functions are employed to convert the gray values in each voxel of the original pictures to a color and opacity value. Afterward, a variety of transfer functions can be used for rendering, based on the case’s features and the structures that should be highlighted. However, the algorithm used in cinematic rendering is based on path-tracing techniques and the global illumination model, which simulate the various paths that billions of photons traveling from all possible directions take through a volumetric dataset and their interaction with the volume to form one pixel. This is in contrast to ray casting techniques, in which each pixel is formed by one light ray. As a result, this recently introduced 3D reconstruction technique provides more realistic 3D images with high spatial resolution and excellent tissue differentiation. Just like VR, CR can offer insights into various tissue types by adjusting the display parameters to enhance the visualization of soft tissues compared to denser structures. CR’s shadowing capabilities can enhance our understanding of anatomy, especially in areas with complex structures, including those with overlapping or protruding elements. However, it is worth noting that the prospective view in CR may partially obscure deeper regions.

In contrast to traditional VR, which treats each reconstructed voxel independently, CR presents neighboring voxels interactively. This adaptability allows for the manipulation of lighting and shadow conditions in a panoramic view. Furthermore, in addition to shadowing, CR’s global lighting model excels at revealing intricate anatomical details, making it particularly useful for visualizing small structures [14]. The use of cinematic rendering in our case allowed a more detailed and accurate visualization of the ectopic teeth in the maxillary sinus. This technique has been shown to improve diagnostic accuracy and reduce the need for additional imaging studies in many clinical conditions [15,16,17,18].

Overall, the use of CT and film rendering in our case allowed us to accurately diagnose and localize the ectopic teeth in the maxillary sinus. These imaging modalities are valuable tools in the diagnosis and management of ectopic teeth in the maxillary sinus and may aid in the planning of surgical interventions [19].

Surgical removal of ectopic teeth in the maxillary sinus is required in symptomatic cases and typically involves extraction of the cyst-associated impacted or unerupted tooth. In some cases, initial marsupialization may be performed to reduce the size of the bony defect prior to enucleation and tooth extraction. The Caldwell–Luc procedure is a common surgical approach for the removal of dentigerous cysts and ectopic teeth in the maxillary sinus. In this procedure, a window is created in the lateral wall of the maxillary sinus, allowing direct visualization and access to the cyst and ectopic tooth. In some cases, endoscopic sinus surgery may be used alone or combined to remove the ectopic tooth, considering that the lateral, anterior, and inferior walls of the maxillary sinus might be challenging to access with the endoscopic technique [20]. In our case, a more conservative approach was preferred, using pharmacological treatment for local inflammation and clinical/imaging follow-up for early detection of possible complications.

## 4. Conclusions

In conclusion, this case report emphasizes the crucial role of advanced imaging techniques such as CT and cinematic rendering in the bilateral maxillary sinus ectopia teeth. Our findings support the potential of CR, using software such as Cinematic Playground, to provide improved visualization of complex anatomical structures and their relationships. This study highlights a rare but clinically significant condition, which can have implications for preoperative planning in cases of maxillary sinus pathology. The literature on this topic is limited, and further studies are required to determine the prevalence of ectopic teeth in the maxillary sinus and their potential impact on surgical outcomes.

## Figures and Tables

**Figure 1 diagnostics-13-03084-f001:**
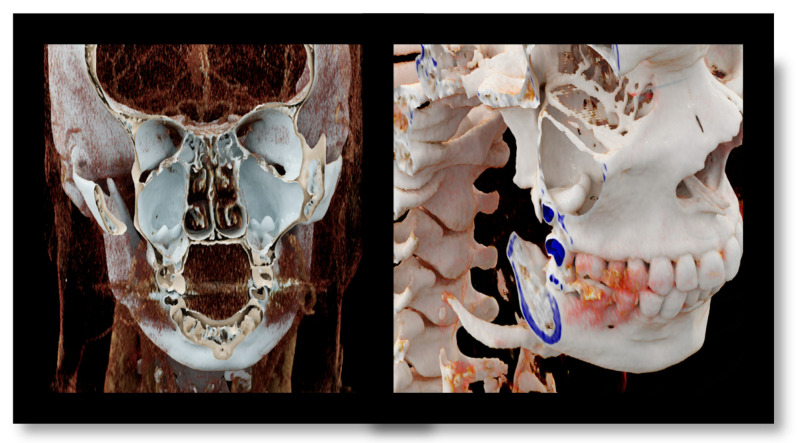
Coronal and para-sagittal sections of the CR showing ectopic teeth in both maxillary sinuses.

**Figure 2 diagnostics-13-03084-f002:**
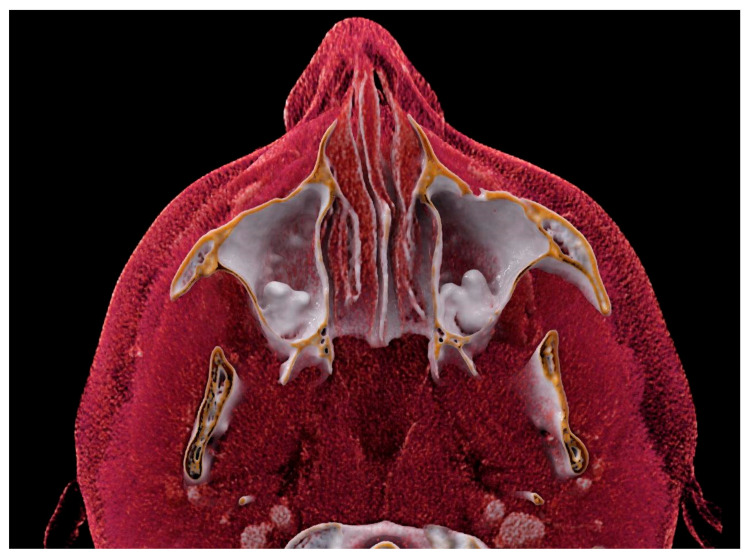
Axial section of the CR showing ectopic teeth in both maxillary sinuses.

**Figure 3 diagnostics-13-03084-f003:**
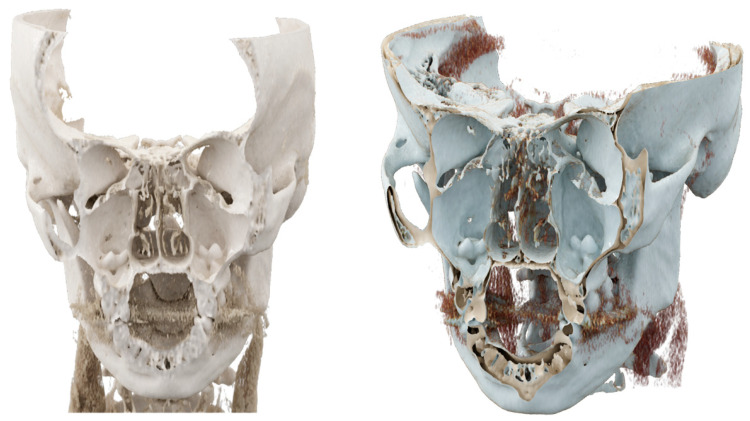
Coronal and para-sagittal sections of the CR showing ectopic teeth in both maxillary sinuses with bone windowing.

## Data Availability

Data is unavailable.

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
