# Peer review of "Evaluation of Bilateral Maxillary Sinus Ectopic Teeth Using CT and Cinematic Rendering—A Case Report"

_diagnostics, 2023, doi:10.3390/diagnostics13193084_

Round 1

Reviewer 1 Report

Dear authors !

I was given the opportunity to review Your manuscript, presenting a case-report of ectopic teeth visualized by cinematic rendering of CAT-scan images.

The case is well presented and all backgrounds of the pathology sufficiently described, pointing out the importance of a proper surgical planning with the best possible visualization by cinematic rendering of CAT-scan images. Maybe, if the journal allows, more figures of the CR-screenshots could be added.

References are up-to-date

The discussion section might be a bit too long and might serve as a distraction of the reader from the main point: the importance of best possible virtual planning by best possible visualization.

Thank You 

Manuscript is written in comprehensible English

Author Response

We thank the reviewer for revisions.

Other images are added to our manuscript for better visualization of ectopic theeth.

Also we have modified the discussion section focusing on the main point: the importance of best possible virtual planning by best possible visualization.

Thank you so much for comments.

Reviewer 2 Report

All the introduction does't show any reference. It must do it.

The last sentence of Discussion tell us that a conservative approach was selected. How long has been this patient followed? It should be posted in order to conclude or not if the conservative approach should be prioritized instead of surgery.

Author Response

Thank you for your thoughtful comment. We would like to clarify that our Institute primarily focuses on diagnostic procedures. We collaborate with specialists to conduct the most appropriate diagnostic tests and provide all possible extracted data. Consequently, our role does not extend to making final treatment decisions or conducting long-term patient follow-up. However, we recognize the importance of thorough follow-up to assess the effectiveness of a conservative approach and will suggest to our clinical colleagues to include such details in future research.

For the first point, we added the requested references on introduction section.

Reviewer 3 Report

The research presents a case of a 28-year-old man with bilateral maxillary sinus ectopic teeth. The topic is relevant and addresses a gap in the field of maxillofacial surgery and orthodontics. It uses a cinematic rendering technique. The authors could present other images of the diagnosis protocol and treatment planning. The conclusion is consistent with the evidence and arguments presented. The references are appropriate. The figures are of high quality, additional ones showing the cinematic rendering technique, treatment planning and diagnosis procedure are welcome.

Could the authors add more data about the performing of cinematic rendering, the used software (including some images) and the treatment of the case?

Author Response

Thank the reviewer for comments.

In the manuscript we added other informations about the performing of cinemating rendering (highlighted in yellow). We added also other cinematic images for better visualization of ectopic theeth.